# RAVE: A VARIATIONAL AUTOENCODER FOR FAST AND HIGH-QUALITY NEURAL AUDIO SYNTHESIS

## ABSTRACT

Deep generative models applied to audio have improved by a large margin the state-of-the-art in many speech and music related tasks. However, as raw waveform modelling remains an inherently difficult task, audio generative models are either computationally intensive, rely on low sampling rates, are complicated to control or restrict the nature of possible signals. Among those models, Variational AutoEncoders (VAE) give control over the generation by exposing latent variables, although they usually suffer from low synthesis quality. In this paper, we introduce a Realtime Audio Variational autoEncoder (RAVE) allowing both fast and high-quality audio waveform synthesis. We introduce a novel two-stage training procedure, namely *representation learning* and *adversarial fine-tuning*. We show that using a post-training analysis of the latent space allows a direct control between the reconstruction fidelity and the representation compactness. By leveraging a multi-band decomposition of the raw waveform, we show that our model is the first able to generate 48kHz audio signals, while simultaneously running 20 times faster than real-time on a standard laptop CPU. We evaluate synthesis quality using both quantitative and qualitative subjective experiments and show the superiority of our approach compared to existing models. Finally, we present applications of our model for *timbre transfer* and *signal compression*. All of our source code and audio examples are publicly available.

## 1 INTRODUCTION

Deep learning applied to audio signals proposes exciting new ways to perform speech generation, musical composition and sound design. Recent works in deep audio modelling have allowed novel types of interaction such as unconditional generation (Chung et al., 2015; Fraccaro et al., 2016; Oord et al., 2016; Vasquez & Lewis, 2019; Dhariwal et al., 2020) or timbre transfer between instruments (Mor et al., 2018). However, these approaches remain computationally intensive, as modeling audio raw waveforms requires dealing with extremely large temporal dimensionality. To cope with this issue, previous approaches usually rely on very low sampling rates (16 to 24kHz), which can be sufficient for speech signals, but is considered as low-quality for musical applications. Furthermore, the auto-regressive sampling procedure used by most models (Engel et al., 2017) is prohibitively long. This precludes real-time application which are pervasive in musical creation, while parallel models (Défossez et al., 2018) can only allow fast generation at the cost of a lower sound quality.

More recently, Engel et al. (2019) and Wang et al. (2019) proposed to leverage classical synthesis techniques to address these limitations, by relying on pre-computed audio descriptors as an extraneous information to condition the models. While these approaches achieve state-of-the-art results in terms of audio quality, naturalness and computational efficiency, the extensive use of audio descriptors highly restricts the type of signals that can be generated. A possible solution to alleviate this issue would be to rely on Variational Autoencoders (Kingma & Welling, 2014), as they provide a form of trainable analysis-synthesis framework (Esling et al., 2018), without explicit restrictions on the type of features learned. However, estimating the dimensionality of the latent representation associated with a given dataset

prior to model training is far from trivial. Indeed, a wrong estimation of the latent dimensionality may result in either poor reconstruction or uninformative latent dimensions, which makes latent exploration and manipulation difficult.

In this paper, we overcome the limitations outlined above by proposing a VAE model built specifically for fast and high-quality audio synthesis. To do so, we introduce a specific two-stage training procedure where the model is first trained as a regular VAE for *representation learning*, then fine-tuned with an *adversarial generation* objective in order to achieve high quality audio synthesis. We combine a multi-band decomposition of the raw waveform alongside classical synthesis blocks inspired by Engel et al. (2019), allowing to achieve high-quality audio synthesis with sampling rates going up to 48kHz without a major increase in computational complexity. We show that our model is able to converge on complex datasets using a low number of parameters, achieving state-of-the-art results in terms of naturalness and audio quality, while being usable in real-time on a standard laptop CPU. We compare our model with several state-of-the-art models and show its superiority in unsupervised audio modeling. In order to address the dimensionality of the learned representation, we introduce a novel method to split the latent space between *informative* and *uninformative* parts using a singular value decomposition, and show that replacing the latter part with random noise does not affect the reconstruction quality. This procedure allows easier exploration and manipulation of latent trajectories, since we only need to operate on a subset of informative latent dimensions. Finally, we discuss the application of our model in *signal compression* and *timbre style transfer*. Our key contributions are:

- A two-stage training procedure where the model is first trained as a regular VAE, then fine-tuned with an adversarial generation objective, as depicted in figure 1
- A post-training analysis of the latent space providing a way to balance between reconstruction fidelity and representation compactness
- High-quality audio synthesis models with sampling rates going up to 48kHz
- 20 times faster than realtime synthesis on a standard laptop CPU

Audio samples and supplementary figures are provided in the accompanying website[1]. We highly encourage readers to listen to accompanying samples while reading the paper.

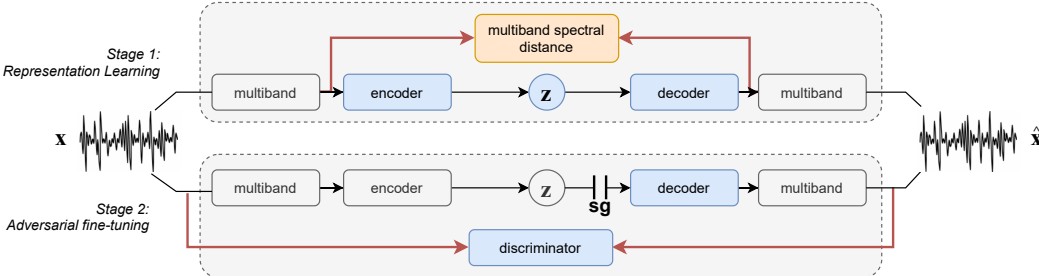

Figure 1: Overall architecture of the proposed approach. Blocks in blue are the only ones optimized, while blocks in grey are fixed or frozen operations.

## 2 STATE-OF-ART

### 2.1 VARIATIONAL AUTOENCODERS

Generative models aim to understand a given dataset $\mathbf{x} \in \mathbb{R}^{d_x}$ by modelling its underlying distribution $p(\mathbf{x})$. To simplify this problem, we can consider that the generation of $\mathbf{x}$ is conditioned by *latent variables* $\mathbf{z} \in \mathbb{R}^{d_z}$, responsible for most of the variations present in $\mathbf{x}$. Therefore, the complete model is defined by the joint distribution $p(\mathbf{x}, \mathbf{z}) = p(\mathbf{x}|\mathbf{z})p(\mathbf{z})$, which is usually not analytically solvable given the complexity of real-world data. Variational

---

[1]https://anonymous84654.github.io/ICLR_anonymous/

autoencoders address this problem by introducing an inference model $q_\phi(\mathbf{z}|\mathbf{x})$, optimized to minimize its Kullback-Leibler (KL) divergence with the true posterior distribution $p(\mathbf{z}|\mathbf{x})$

$$\phi^* = \underset{\phi}{\arg\min} \ \ \mathcal{D}_{\mathrm{KL}}[q_\phi(\mathbf{z}|\mathbf{x})\|p(\mathbf{z}|\mathbf{x})], \tag{1}$$

which can be rearranged to obtain the final objective used to train a VAE, called the Evidence Lower BOund (ELBO), as shown by Kingma & Welling (2014)

$$\mathcal{L}_{\phi,\theta}(\mathbf{x}) = -\mathbb{E}_{q_\phi(\mathbf{z}|\mathbf{x})}[\log p_\theta(\mathbf{x}|\mathbf{z})] + \mathcal{D}_{\mathrm{KL}}[q_\phi(\mathbf{z}|\mathbf{x})\|p(\mathbf{z})]. \tag{2}$$

The ELBO minimizes the reconstruction error of the model through the likelihood of the data given a latent $\log p_\theta(\mathbf{x}|\mathbf{z})$, while regularizing the posterior distribution $q_\phi(\mathbf{z}|\mathbf{x})$ to match a predefined prior $p(\mathbf{z})$. Both posterior distributions $q_\phi$ and $p_\theta$ are parametrized by neural networks respectively called *encoder* and *decoder*. Higgins et al. (2016) proposes to weight the KL divergence in equation (2) with a parameter $\beta$ to control the trade-off between accurate reconstruction and strong latent regularization. They show that increasing $\beta > 1$ leads to less entangled latent dimensions, at the detriment of the reconstruction quality.

## 2.2 Autoencoding raw waveform

One of the first approaches addressing the raw waveform modelling task are WaveNet (Oord et al., 2016) and SampleRNN (Mehri et al., 2017), where the probability of a waveform $\mathbf{x}$ is factorized as a product of conditional probabilities

$$p(\mathbf{x}) = \prod_{t>1} p(x_t|x_1, \ldots, x_{t-1}). \tag{3}$$

Those models require a large amount of data and parameters to properly converge. Furthermore, the autoregressive nature of the synthesis process makes it prohibitively slow, and prone to accumulate errors.

WaveNet has also been adapted by Engel et al. (2017) for their NSynth model, addressing the representation learning task. Unlike equation (2), they do not regularize the learned representation, and rather encode the raw waveform deterministically to its latent counterpart. It implies the absence of a prior distribution $p(\mathbf{z})$ and, therefore, prevents sampling from the latent space. This restricts the applications of the model to simple reconstructions and interpolations.

As a way to speed-up the synthesis process, Défossez et al. (2018) proposed an autoencoder with feed-forward convolutional networks parametrizing both the encoder and the decoder. They use a perceptually-motivated distance between waveforms called *spectral distance* as the reconstruction objective

$$l(\mathbf{x}, \mathbf{y}) = \left\|\log(\mathrm{STFT}(\mathbf{x})^2 + \epsilon) - \log(\mathrm{STFT}(\mathbf{y})^2 + \epsilon)\right\|_1, \tag{4}$$

where STFT is the Short-Term Fourier Transform. Since they use a squared STFT, the phase component is discarded making the loss permissive to inaudible phase variations. They show that their model is 2500 times faster that *NSynth* during synthesis, at the expense of a degraded sound quality.

Following the recent advances in generative adversarial modelling (Goodfellow et al., 2014), Kumar et al. (2019) proposed to use an adversarial objective to address the parallel audio modelling task. The discriminator is trained to differentiate true samples from generated ones, while the generator is optimized to produce samples that are classified as true by the discriminator. A *feature matching loss* is added to the adversarial loss, minimizing the L1 distance between the discriminator feature maps of real and synthesized audio. This feature matching mechanism can be seen as a learned metric to evaluate the distance between two samples, and has been successfully applied to the conditional waveform modelling task (e.g spectrogram to waveform or replacement of the decoder in a pretrained autoencoder model).

# 3 METHOD

## 3.1 TWO-STAGE TRAINING PROCEDURE

Ideally, the representation learned by a variational autoencoder should contain *high-level* attributes of the dataset. However, two perceptually similar audio signals may contain subtle phase variations that produce dramatically different waveforms. Hence, estimating the reconstruction term in equation (2) using the raw waveform penalizes the model if those subtle variations are not included in the learned representation. This might both hamper the learning process and include in the latent space those *low-level* variations about audio signal that are not relevant perceptually. To address this problem, we split the training process in two stages, namely *representation learning* and *adversarial fine-tuning*.

### 3.1.1 STAGE 1: REPRESENTATION LEARNING

The first stage of our procedure aims to perform *representation learning*. We leverage the multiscale spectral distance $S(\cdot, \cdot)$ proposed by Engel et al. (2019) in order to estimate the distance between real and synthesized waveforms, defined as

$$S(\mathbf{x}, \mathbf{y}) = \sum_{n \in \mathcal{N}} \left[ \frac{\|\mathrm{STFT}_n(\mathbf{x}) - \mathrm{STFT}_n(\mathbf{y})\|_F}{\|\mathrm{STFT}_n(\mathbf{x})\|_F} + \log\left(\|\mathrm{STFT}_n(\mathbf{x}) - \mathrm{STFT}_n(\mathbf{y})\|_1\right) \right], \quad (5)$$

where $\mathcal{N}$ is a set of scales, $\mathrm{STFT}_n$ is the amplitude of the Short-Term Fourier Transform with window size $n$ and hop size $n/4$, and $\|\cdot\|_F$, $\|\cdot\|_1$ are respectively the Frobenius norm and $L_1$ norm. Using an amplitude spectrum-based distance does not penalize the model for inaccurately reconstructed phase, but encompasses important perceptual features about the signal. We train the *encoder* and *decoder* with the following loss derived from the ELBO

$$\mathcal{L}_{\mathrm{vae}}(\mathbf{x}) = \mathbb{E}_{\hat{\mathbf{x}} \sim p(\mathbf{x}|\mathbf{z})}[S(\mathbf{x}, \hat{\mathbf{x}})] + \beta \times \mathcal{D}_{\mathrm{KL}}[q_\phi(\mathbf{z}|\mathbf{x})\|p(\mathbf{z})], \quad (6)$$

We start by training the model solely with $\mathcal{L}_{\mathrm{vae}}$, and once this loss converges, we switch to the next training phase.

### 3.1.2 STAGE 2: ADVERSARIAL FINE-TUNING

The second training stage aims at improving the synthesized audio quality and naturalness. As we consider that the learned representation has reached a satisfactory state at this point, we freeze the encoder and only train the decoder using an adversarial objective.

GANs are *implicit* generative models allowing to sample from a complex distribution by transforming a simpler one, called the *base distribution*. Here, we use the learned latent space in the first stage as the base distribution, and train the decoder to produce synthesized signals similar to the real ones by relying on a *discriminator D*. We use the hinge loss version of the GAN objective, defined as

$$\mathcal{L}_{\mathrm{dis}}(\mathbf{x}, \mathbf{z}) = \max(0, 1 - D(\mathbf{x})) + \mathbb{E}_{\hat{\mathbf{x}} \sim p(\mathbf{x}|\mathbf{z})}[\max(0, 1 + D(\hat{\mathbf{x}}))],$$
$$\mathcal{L}_{\mathrm{gen}}(\mathbf{z}) = -\mathbb{E}_{\hat{\mathbf{x}} \sim p(\mathbf{x}|\mathbf{z})}[D(\hat{\mathbf{x}})]. \quad (7)$$

In order to ensure that the synthesized signal $\hat{\mathbf{x}}$ does not diverge too much from the ground truth $\mathbf{x}$, we keep minimizing the spectral distance defined in equation (5), but also add the feature matching loss $\mathcal{L}_{\mathrm{FM}}$ proposed by Kumar et al. (2019). Altogether, this yields the following objective for the decoder

$$\mathcal{L}_{\mathrm{total}}(\mathbf{x}, \mathbf{z}) = \mathcal{L}_{\mathrm{gen}}(\mathbf{z}) + \mathbb{E}_{\hat{\mathbf{x}} \sim p(\mathbf{x}|\mathbf{z})}[S(\mathbf{x}, \hat{\mathbf{x}}) + \mathcal{L}_{\mathrm{FM}}(\mathbf{x}, \hat{\mathbf{x}})]. \quad (8)$$

## 3.2 Latent representation compactness

The loss proposed in equation (6) contains two terms, a *reconstruction* and *regularisation* term. Those two terms are somewhat conflicting, since the reconstruction term maximises the mutual information between the latent representation and the data distribution, while the regularisation term guides the posterior distribution towards independence with the data (potentially causing *posterior collapse*). In practice, the pressure applied by the regularisation term to the encoder during training encourages it to learn a compact representation, where informative latents have the highest KL divergence from the prior, while uninformative latents have a KL divergence close to 0 (Higgins et al., 2016).

Here, we address the task of identifying the most informative parts of the latent space in order to restrict the dimensionality of the learned representation to the strict minimum required to reconstruct a signal. To do so, we adapt the method for range and null space estimation (see appendix A) to this problem. Let $\mathbf{Z} \in \mathbb{R}^{b \times d}$ be a matrix composed of $b$ samples $\mathbf{z} \in \mathbb{R}^d$, where $\mathbf{z} \sim q_\phi(\mathbf{z}|\mathbf{x})$. Using a Singular Value Decomposition (SVD) directly on $\mathbf{Z}$ to solve the problem of finding informative parts of the latent space would not be relevant given the high variance present in the collapsed parts of $\mathbf{Z}$. In order to adapt this to our problem, we first remove the variance from $\mathbf{Z}$, by considering the matrix $\mathbf{Z}' \in \mathbb{R}^{b \times d}$ that verifies

$$\mathbf{Z}'_i = \operatorname*{argmax}_{\mathbf{z}} \ q_\phi(\mathbf{z}|\mathbf{x}), \tag{9}$$

Hence, dimensions of the posterior distribution $q_\phi(\mathbf{z}|\mathbf{x})$ that have collapsed to the prior $p(\mathbf{z})$ will result in a constant value in $\mathbf{Z}'$, that we set to 0 by removing the average of $\mathbf{Z}'$ across the first dimension. The only dimensions of $\mathbf{Z}'$ with non-zero values are therefore correlated with the input, which constitute the informative part of the latent space. Applying a SVD on this centered matrix, we can obtain the matrix $\mathbf{\Sigma}$ containing the singular values of $\mathbf{Z}'$, by computing

$$\mathbf{Z}' = \mathbf{U}\mathbf{\Sigma}\mathbf{V^T}, \tag{10}$$

As detailed in appendix A, the rank $r$ of $\mathbf{Z}'$ is equal to the number of non-zero singular values in $\mathbf{\Sigma}$. Given the high variation that exists in real-world data, it is unlikely that the vanishing singular values of $\mathbf{Z}'$ are equal to 0. Therefore, instead of tracking the exact rank $r$ of $\mathbf{Z}'$, we define a *fidelity* parameter $f \in [0-1]$, with the associated rank $r_f$ defined as the smallest integer verifying

$$\frac{\sum_{i \leq r_f} \mathbf{\Sigma}_{ii}}{\sum_i \mathbf{\Sigma}_{ii}} \geq f. \tag{11}$$

Given the fidelity value $f$, and a latent representation $\mathbf{z} \sim q_\phi(\mathbf{z}|\mathbf{x})$, we reduce the dimensionality of $\mathbf{z}$ by projecting it on the basis defined by $\mathbf{V}^T$ and keep only the $r_f$ first dimensions. We obtain a low-rank representation $\mathbf{z}_f$, whose dimensionality depends on both the dataset and $f$. Before providing $\mathbf{z}_f$ to the decoder, we concatenate it with noise sampled from the prior distribution, and project it back on its original basis using $\mathbf{V}$. We demonstrate in section 5.3 the influence of $f$ on the reconstruction.

## 4 Experiments

### 4.1 RAVE

Here, we introduce our Realtime Audio Variational autoEncoder (RAVE) built for high-quality audio representation learning and faster than realtime synthesis. Since we target the modelling of 48kHz audio signals, we leverage a multiband decomposition of the raw waveform (see appendix B) as a way to decrease the temporal dimensionality of the data (Yang et al., 2020). This allows us to expand the temporal receptive field of our model

without a major increase in computational complexity. We demonstrate the influence of the multiband decomposition on the synthesis speed in section 5.2.

Using a 16-band decomposition, we successfully model 48kHz audio signals while producing a compact latent representation using the post-training analysis presented in section 3.2.

**Encoder.** We define our encoder as the combination of a multiband decomposition followed by a simple convolutional neural network, transforming the raw waveform into a 128-dimensional latent representation. A detailed description of the architecture of the encoder is given in annex C.1.

**Decoder.** Our decoder is a modified version of the generator proposed by Kumar et al. (2019). We use the same alternation of upsampling layers and residual networks, but instead of directly outputting the raw waveform we feed the last hidden layer to three sub-networks. The first sub-network (*waveform*) synthesizes a multiband audio signal (with *tanh* activation), which is multiplied by the output of the second sub-network (*loudness*), generating an amplitude envelope (with *sigmoid* activation). The last sub-network is a noise synthesizer as proposed in Engel et al. (2019), and produces a multiband filtered noise added to the previous signal.

We found that using an explicit amplitude envelope helps reducing artifacts in the silent parts of the signal, while the noise synthesizer slightly increases the reconstruction naturalness of noisy signals. See annex C.2 for more details about the architecture of the decoder.

**Discriminator.** We use the exact same discriminator as in Kumar et al. (2019), which is a strided convolutional network applied on different scales of the audio signal to prevent artifacts. We also use the same feature matching loss as in the original paper.

**Training.** We follow the training procedure proposed in 3.1, and train RAVE for 3M steps, specifically 1.5M steps for each stage, summing to a total of approximately 6 days on a single TITAN V GPU. We use the *Adam* optimizer (Kingma & Ba, 2015) with a learning rate of $10^{-4}$, $\beta = (0.5, 0.9)$, and batch size 8. We use dequantization, random crop and allpass filters with random coefficients as our data augmentation strategy.

**Baselines.** We evaluate our model in the context of unsupervised representation learning. We compare it against two state-of-the-art models: an unsupervised NSynth (Engel et al., 2017) and the autoencoder from SING (Défossez et al., 2018). We use the official implementations for both baselines with default parameters. We do not to evaluate the DDSP (Engel et al., 2019) and NSF (Wang et al., 2019) approaches since we target the unsupervised modelling of any type of audio signals. Notably, our proposal is able to model both monophonic and polyphonic signals, while the aforementioned methods are restricted to monophonic signals.

### 4.2 DATASETS

**Strings.** Since our main target is the modelling of musical audio signals, we use an internal dataset composed of approximately 30 hours of raw recordings of strings in various configurations (monophonic solos and polyphonic group performances, with different styles and recording configurations) sampled at 48kHz. We employ a 90/10 train/test split. We downsample this dataset to 16kHz when using NSynth and SING.

**VCTK.** The Voice Conversion ToolKit (Yamagishi et al., 2019) is a speech dataset composed of approximately 44 hours of raw audio sampled at 48kHz, produced by 110 different speakers with various accents. We use it to evaluate the performances of RAVE when addressing the speech modelling task in an unsupervised fashion. We also employ a 90/10 train/test split and downsample to 16 kHz for other methods.

## 5 RESULTS

To encourage reproducibility, we release the source code of our model alongside pretrained weights used to produce the presented results. Audio samples corresponding to the different parts of this section are available on our accompanying website.

### 5.1 SYNTHESIS QUALITY

First, we performed a qualitative experiment where participants were asked to rate audio samples on a scale of 1 to 5. The test consisted of 15 trials, with each presenting an audio sample of the *strings* dataset never seen during training, alongside its reconstruction by the three models. The order in which the trials and samples are presented was randomized. A total of 33 participants took the test, with most being audio professionals. We report the results of this study in table 1.

Table 1: Reconstruction quality evaluation (Mean Opinion Score)

| Model | MOS | 95% CI | Training time | Parameter count |
|---|---|---|---|---|
| Ground truth | 4.21 | ±0.04 | - | - |
| NSynth | 2.68 | ±0.04 | ∼ 13 days | 64.7M |
| SING | 1.15 | ±0.02 | **∼ 5 days** | 80.8M |
| **RAVE (Ours)** | **3.01** | ±0.05 | ∼ 7 days | **17.6M** |

As we can see, RAVE outperforms both NSynth and SING in terms of audio quality without relying on autoregressive generation. Furthermore, it achieves high-quality audio synthesis with at least 3.5 times less parameters. There is still a gap in the evaluation between the ground truth and the other models. This might come from the difficulty of modeling the variety of room acoustics conditions present in the original dataset (as discussed in Engel et al. (2019)), sometimes making it obvious that the evaluated samples are synthesized.

### 5.2 SYNTHESIS SPEED

Several approaches using normalizing flows (Prenger et al., 2019) or adversarial models (Kumar et al., 2019) can achieve faster than realtime synthesis, but only by relying on GPUs. Other approaches make strong assumptions about the signal in order to simplify the generation process (Engel et al., 2019; Wang et al., 2019), allowing real-time synthesis on CPU, while restricting the range of audio signals that can be modeled. In table 2, we evaluate the synthesis speed of all models on both CPU and GPU. The synthesis speed is calculated as the average number of audio samples generated per second for 100 trials.

Table 2: Comparison of the synthesis speed for several models

| Model | CPU synthesis | GPU synthesis |
|---|---|---|
| NSynth | 18 Hz | 57 Hz |
| SING | 304 kHz | 9.8 MHz |
| RAVE (Ours) w/o multiband | 38 kHz | 3.7 MHz |
| **RAVE (Ours)** | **985 kHz** | **11.7 MHz** |

Being the only model relying on autoregressive synthesis, NSynth is also the slowest, peaking at 57Hz during generation. As expected, the parallel nature of SING and RAVE makes them several orders of magnitude faster than NSynth. The addition of the multiband decomposition speeds up RAVE by a factor of 25, allowing our model to outperform SING on both CPU and GPU. Overall, we obtain audio synthesis at 48kHz with a 20× faster than realtime factor on CPU, and up to 240× on GPU.

## 5.3 Balancing compactness and fidelity

Having a compact learned representation has several benefits, since it is easier to analyse, manipulate and understand. However, it usually comes at the expense of trading off the reconstruction quality. Instead of determining the latent space dimensionality prior to the training, we rely on the fidelity parameter $f$ to estimate it post-training and accordingly crop the learned representation, as explained in section 3.2.

In figure 2, we compute the relationship between $f$ and the estimated number of dimensions $r_f$ for both the *strings* and *vctk* datasets. We depict the influence of $f$ on the reconstruction quality by measuring the spectral distance (see equation 5) between the original and reconstructed samples.

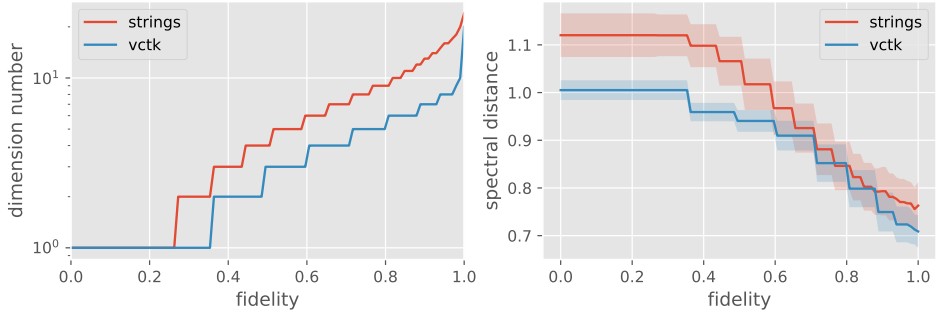

Figure 2: Estimated latent space dimensionality according to the fidelity parameter and its corresponding influence on the reconstruction quality.

By setting $f = 0.99$, the dimensionality of the learned representation drops from 128 to just 24 on the *strings* dataset and 16 on the *vctk* dataset. The downsampling factor of the encoder is 2048, resulting in a latent representation sampled at $\sim 23$Hz. Further decreasing $f$ results in a higher spectral distance as shown in figure 2 (right), but significantly reduces the learned representation size. To exemplify this, we reconstruct a sample from the *vctk* dataset using different fidelity values, and display the resulting melspectrograms in figure 3.

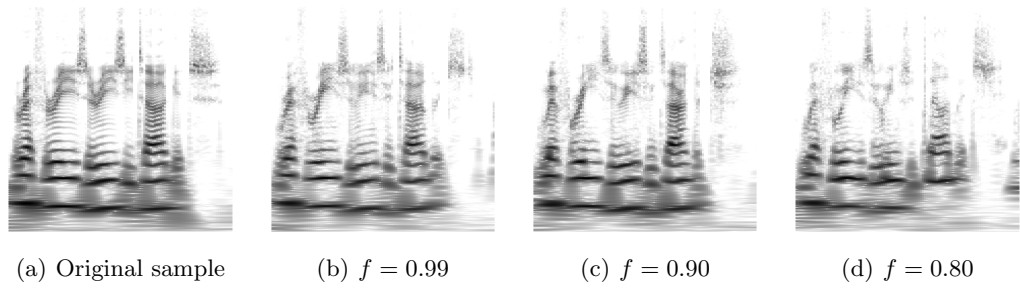

(a) Original sample     (b) $f = 0.99$     (c) $f = 0.90$     (d) $f = 0.80$

Figure 3: Reconstruction of an input sample with several fidelity parameters $f$.

As the fidelity parameter decreases, reconstructed samples get less accurate, loosing parts of their attributes such as phonemes or speaker identity.

## 5.4 Timbre transfer

We demonstrate that RAVE can be used to perform domain transfer even if it has not been specifically trained to address this particular task. We perform domain transfer by simply providing to a pretrained RAVE model audio samples coming from outside of its original training distribution (e.g violin samples are reconstructed with a model trained on speech, see section F for more details).

### 5.5 Signal compression

Since the representation learned by RAVE is significantly more compact than the raw audio waveform, it can be used as a data-driven compression system for transmission and storage purposes, or to produce the base signal for a higher-level model to work on. Applying our model provides a compression ratio of 2048, producing latent signals sampled at $\sim 23$Hz, which can be used as a simpler representation for further learning tasks.

To illustrate this, we train a WaveNet-inspired model at generating latent signals in an autoregressive fashion for performing audio synthesis with RAVE. Combining this autoregressive model with the decoder from RAVE, we still obtain synthesis at a 5 times faster than realtime factor. Furthermore, since latent signals are sampled at a much slower rate than the raw waveform, we obtain a 3-second-long receptive field with as few as 9M parameters, corresponding to only 10 layers of the original WaveNet architecture. Examples of unconditional generation are available on our accompanying website.

## 6 Related work

The combination of VAE and GAN has already been studied as a way to build a representation learning model with high quality generation abilities. Larsen et al. (2015) propose to replace the reconstruction loss with a learned metric similar to the feature matching loss we use in equation 8. This learned metric can be seen as a perceptual loss (Kumar et al., 2019), but we demonstrate in annex E that using it to train our encoder results in a larger estimated latent space dimensionality, making the learned representation harder to analyse and interact with.

Previous approaches combine perceptual losses and auxiliary losses to achieve high quality audio synthesis. Yamamoto et al. (2020) leverage a multiscale spectral loss together with an adversarial objective to address the mel-scale spectrogram inversion task, while Ping et al. (2018) use a spectral loss as a guide to stabilize their distillation procedure. Contrary to our two-stage procedure, they mainly use this combination as a warmup mechanism to help their model converge, whereas we use both losses with two separate tasks in mind: *representation learning* encompassing spectral attributes of the signal, and *high-quality audio synthesis* based on an adversarial objective, using a fixed learned latent space.

The use of a multiband decomposition of the raw waveform has been successfully applied by Yu et al. (2019) and Yang et al. (2020) to the audio modelling task, and they both show that it helps producing higher quality results while reducing the training and synthesis time. Modelling 48kHz audio is challenging in term of temporal complexity and memory requirements, and building up from the aforementioned work we show that using a 16 band decomposition allows RAVE to address this task.

## 7 Conclusion

In this paper, we introduced RAVE: a Realtime Audio Variational autoEncoder for fast and high-quality neural audio synthesis. First, we proposed a two-stage training procedure, which ensures that the latent representation is adequate, prior to performing adversarial fine-tuning for generating high-quality audio signals. We showed that our method outperforms the performances of previous approaches in both quantitative and qualitative analyses. By leveraging a multiband decomposition of the raw waveform, we are able to achieve 20 times faster than realtime synthesis on a standard laptop CPU. Finally, we proposed a method to control the trade-off between reconstruction quality and representation compactness, easing the analysis and control of the learned representation. We provide the open-source code of RAVE and hope that this will spark contributions and creative uses of our model.

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

## A  RANGE AND NULL SPACE ESTIMATION

Let $\mathbf{Z}$ be a $b \times d$ real-valued matrix with $\sum_i Z_{ij} = 0$. We consider the singular value decomposition of $\mathbf{Z}$

$$\mathbf{Z} = \mathbf{U\Sigma V^T} \tag{12}$$

where $\mathbf{U}$ and $\mathbf{V}$ are orthogonal real-valued matrices of respective dimensions $b \times b$ and $d \times d$, and $\mathbf{\Sigma}$ is a rectangular diagonal matrix with non-negative values on the diagonal called *singular values*, sorted by decreasing value. The number of non-zero singular values gives the *rank* $r$ of the matrix, i.e the dimension of the vector space spanned by its columns.

The range and null space associated with $\mathbf{Z}$ are respectively spanned by the first $r$ and last $d - r$ columns of $\mathbf{V^T}$. A low rank version of $\mathbf{Z}$ can be obtained by keeping only the $r$ first columns of $\mathbf{\Sigma}$ and $\mathbf{V}$, such that

$$\mathbf{Z} = \mathbf{U\tilde{\Sigma}\tilde{V}^T}, \tag{13}$$

where $\mathbf{\tilde{\Sigma}}$ and $\mathbf{\tilde{V}^T}$ are respectively of dimension $b \times r$ and $r \times d$. Note that the equality in equation (13) only holds if the last $d - r$ singular values are equal to 0. Rearranging equation (13), we get the formula to perform a Principal Component Analysis (PCA) on $\mathbf{Z}$ with $r$ components

$$\mathbf{\tilde{Z}} = \mathbf{Z\tilde{V}} = \mathbf{U\tilde{\Sigma}}. \tag{14}$$

Since $\mathbf{V}$ is orthogonal, we can get $\mathbf{Z}$ back by multiplying equation (14) by $\mathbf{V^T}$.

## B  MULTIBAND DECOMPOSITION

The main use of a multiband decomposition is to represent an audio signal sampled at a given sampling rate (e.g 48kHz) as a combination of several downsampled sub-signals (e.g 3kHz), where each sub-signal covers a particular range of frequencies. This decomposition is useful for compression purposes such as mp3 encoding/decoding, since it allows the use of different bit rates on specific areas of the spectrum.

Multiband decompositions of the raw waveform have already been repeatedly applied in the context of audio generative modelling. (Yu et al., 2019) specifically use Pseudo Quadrature Mirror filters (PQMF), defined by Nguyen (1994). PQMF are $M-$band filter-banks that split a signal into $M$ sub-signals decimated by a factor $M$. The $M$ filters are cosine modulations of a prototype low-pass filter with cutoff frequency

$$f_c = \frac{\text{sr}}{2M}, \tag{15}$$

carefully designed to avoid aliasing in the reconstructed signal. In practice, it is impossible to create a perfect prototype filter (i.e perfectly rejected band), but we can instead design a filter that allows the cancellation of aliasing between neighbouring sub-bands, using methods such as the optimisation of a non-linear objective function as described by M. Rossi et al. (1996), or using a simple Kaiser window whose parameters are optimised in an analysis-synthesis pipeline, as proposed by Lin & Vaidyanathan (1998).

Since the filter-bank is an orthogonal basis of the signal due to the cosine modulations implied during its creation, the re-synthesis process can be performed by reapplying a temporally flipped version of the filter-bank to the sub-bands.

## C MODEL ARCHITECTURE

### C.1 ENCODER

The encoder of RAVE is a convolutional neural network with leaky ReLU activation and batch normalization (see figure 4). For all of our experiments we use $N = 4$, with hidden sizes $[64, 128, 256, 512]$ and strides $[4, 4, 4, 2]$. The full latent space has 128 dimensions.

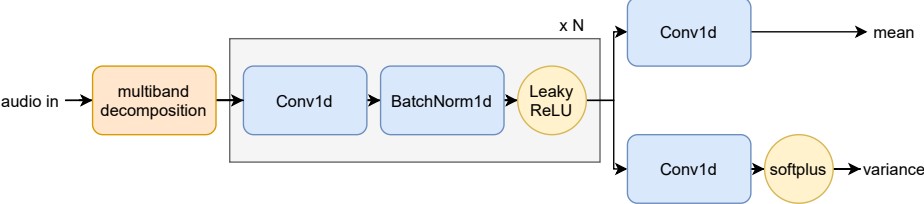

Figure 4: Architecture of the encoder used in the RAVE model.

### C.2 DECODER

We detail the different blocks composing the decoder.

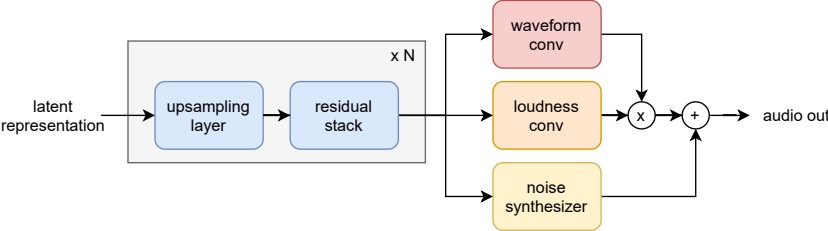

Figure 5: Overview of the proposed decoder. The latent representation is upsampled using alternating upsampling layers and residual stack. The result is processed by three sub-networks, respectively producing *waveform*, *loudness* envelope and filtered *noise* signals.

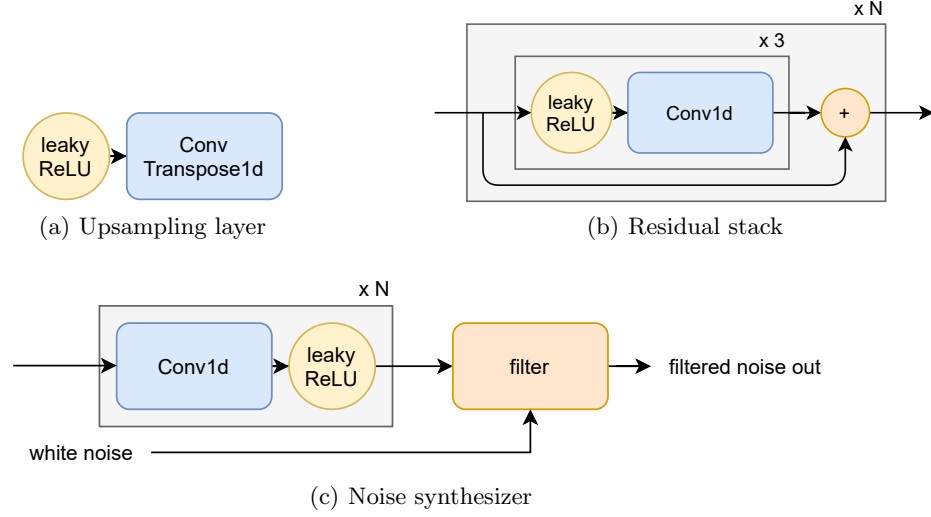

Figure 6: Detailed architecture of the decoder blocks used in the RAVE model.

# D  LATENT COMPONENT COLLAPSE

The regularisation term in equation 6 applies a pressure on the encoder to produce a posterior distribution that is close to the prior, and therefore not correlated with the input data. Since this objective is contrary to the reconstruction objective, it has been empirically shown by Higgins et al. (2016) that some latents are more informative than others, as it can be seen in figure 7 in the case of RAVE.

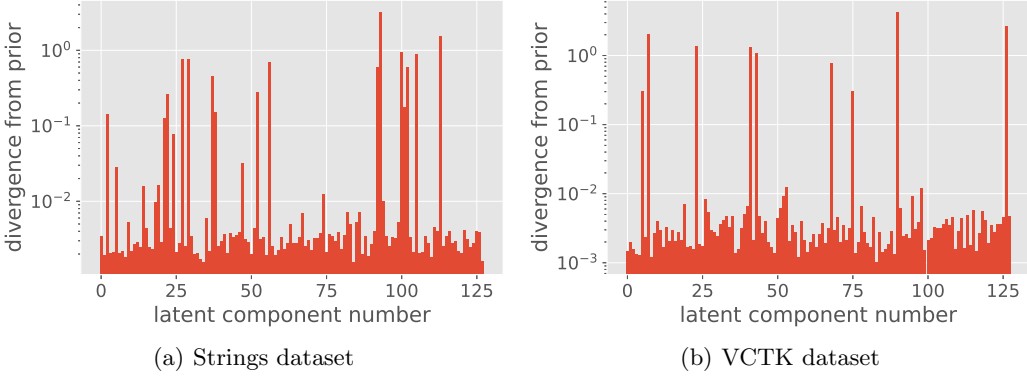

(a) Strings dataset  (b) VCTK dataset

Figure 7: Mean Kullback Leibler divergence for each latent component between the posterior distribution $q_\phi(\mathbf{z}|\mathbf{x})$ estimated over the test set of the strings and VCTK datasets and the prior distribution.

For both datasets, most of the latents have a low KL divergence (lower than 0.01), and only a few (respectively 16 and 9 for the strings and VCTK datasets) have a KL divergence higher than 0.1, which is consistent with the dimensionality estimated by the method proposed in section 3.2 for a fidelity parameter $f = 0.95$.

# E  TWO STAGE TRAINING AND LATENT SPACE COMPACTNESS

Compared to previous approaches combining VAE and GANs, there are no adversarial losses involved when training the encoder. We demonstrate in figure 8 how training the encoder to minimize the feature matching loss as proposed by Larsen et al. (2015) during the second stage results in a dramatically increased estimated latent space dimensionality, which goes against our objective of building a compact representation.

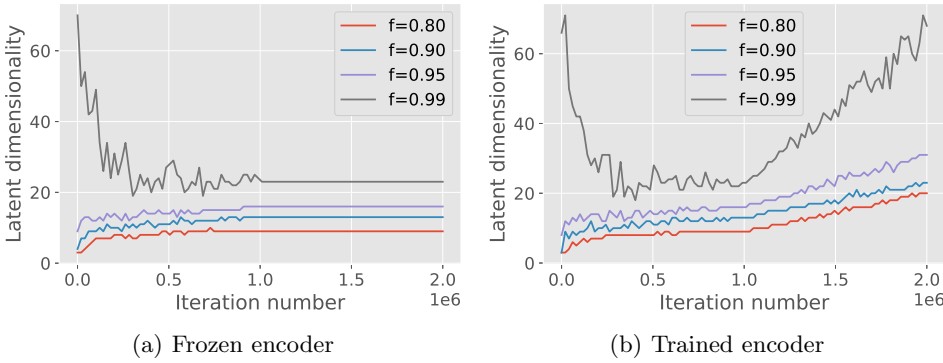

(a) Frozen encoder  (b) Trained encoder

Figure 8: Comparison of the estimated latent space dimensionality for two trainings of RAVE on the Strings dataset with and without freezing the encoder during the *adversarial fine tuning* stage (starting at $1.10^6$ iterations).

## F    Out of domain latent representation

We demonstrate in figure 9 that RAVE can be used to address the timbre transfer task (see section 5.4).

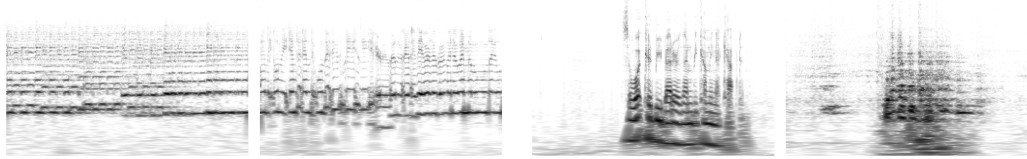

(a) Transfer from *strings* to *vctk*               (b) Transfer from *vctk* to *strings*

Figure 9: Example of timbre transfer using RAVE.

High-level attributes such as the overall loudness and the fundamental frequency of the harmonic components are kept after performing domain transfer. Other audio attributes such as formants are absent in the examples coming from the *strings* dataset, and are added by the model in the speech-transfered version.

However, there are no guarantees that the encoder will encode signals from an out-of-domain distribution into a latent representation that matches the prior. We empirically evaluate the KL divergence of two trained RAVE models for in and out-of-domain data distributions, and report the results in table 3.

Table 3: KL divergence from the prior for in and out-of-domain data.

|  | RAVE Strings | RAVE VCTK |
|---|---|---|
| data from Strings | $\mathbf{0.11 \pm 0.09}$ | $0.16 \pm 0.11$ |
| data from VCTK | $0.24 \pm 0.37$ | $\mathbf{0.09 \pm 0.02}$ |

Unsurprisingly, the smallest KL divergences are observed when encoding data sampled from the training distribution, while encoded out-of-domain signals roughly double the KL divergence. Stronger transfer abilities may be achieved by integrating a domain adaptation technique such as the one proposed by Lample et al. (2017) or Mor et al. (2018).

