# OpenReview forum: "RAVE: A variational autoencoder for fast and high-quality neural audio synthesis"
_ICLR.cc/2022/Conference — ICLR 2022 Submitted_

### Official Review · Reviewer_4x1t · 2021-11-02

**Correctness:** 3
**Technical Novelty And Significance:** 2
**Empirical Novelty And Significance:** 2
**Recommendation:** 5
**Confidence:** 3

**Main Review:**

– Missing references to Fraccaro et al, 2016, and Chung et al, 2016 with the SRNN and VRNN respectively.
– Missing references to other literature using VAE and GAN in combination for training.
– Eq. 2 is only the real ELBO for beta=1. That should be stated.
– Section 3.2 paragraph 1: You would need some references to underline these claims. The reviewer would argue that these are to a certain extent not always true when ref to literature.
– It seems to me that the main evaluation of the model is performed in Table 1. However, these samples are reconstructions. How can you make sure that after fine-tuning the model, your KL-term remains low so that your sample quality can be high?
– In the article, you claim quantitative evaluation of the proposed model. Since you have an ELBO in the model, it would be very nice to see an evaluation of the approximation to the likelihood and compare this to other models, e.g., a purely autoregressive model with its exact likelihood.

**Summary Of The Paper:**

The authors propose a model for audio generation, trained in two stages: 1) as a changed VAE with a likelihood evaluated by comparing the STFT representations and a temperature (beta) on the KL term in order to avoid posterior collapse, 2) as a GAN-optimization scheme to fine-tune the model.

**Summary Of The Review:**

Thank you for a very nice read. While finding the results interesting, I have a hard time arguing whether the proposed innovations are in fact impactful. I'm lacking quantitative experiments, e.g., ablations studies, on the components and the optimization details introduced.

---

> ### Author Response · Authors · 2021-11-17
> **Response to Reviewer 4x1t**
>
> Thank you for your review and your encouraging comments !
>
> > “The authors propose a model for audio generation, trained in two stages: 1) as a changed VAE with a likelihood evaluated by comparing the STFT representations and a temperature (beta) on the KL term in order to avoid posterior collapse, 2) as a GAN-optimization scheme to fine-tune the model.”
>
> As we note in Reviewer Pcii’s comment, we also propose a way to analyse the latent space post training, and give a way to trade fidelity for representation compactness.
>
> > “Missing references to Fraccaro et al, 2016, and Chung et al, 2016 with the SRNN and VRNN respectively.”
>
> Agreed, we have added those references.
>
> > “Missing references to other literature using VAE and GAN in combination for training.”
>
> Agreed, we have added a related work section where we discuss those approaches.
>
> > “Eq. 2 is only the real ELBO for beta=1. That should be stated”
>
> Indeed ! We changed it.
>
> > “Section 3.2 paragraph 1: You would need some references to underline these claims. The reviewer would argue that these are to a certain extent not always true when ref to literature.”
>
> Agreed ! We have reformulated our claim and added a reference. We have also added an experiment where we show the KL divergence for each component of the latent space average over the entire dataset.
>
> > “ It seems to me that the main evaluation of the model is performed in Table 1. However, these samples are reconstructions. How can you make sure that after fine-tuning the model, your KL-term remains low so that your sample quality can be high?”
>
> During the fine tuning of our model, the encoder is frozen, hence the KL divergence of the posterior distribution from the prior is not affected.
>
> > “In the article, you claim quantitative evaluation of the proposed model. Since you have an ELBO in the model, it would be very nice to see an evaluation of the approximation to the likelihood and compare this to other models”
>
> It would have been very interesting, however we do not use an ELBO in our model, since we essentially replaced the expectation over the posterior of the data’s log probability given the latent with a spectral distance, preventing any likelihood approximation.
>
> > “While finding the results interesting, I have a hard time arguing whether the proposed innovations are in fact impactful.”
>
> We hope that our updated version of the article will make clear how our approach is different from previous work.
>
> > “I'm lacking quantitative experiments, e.g., ablations studies, on the components and the optimization details introduced.”
>
> We have added an experiment to the annex showing how the two-stage training is crucial to maintain a compact learned representation.

---

### Official Review · Reviewer_Pcii · 2021-11-03

**Correctness:** 2
**Technical Novelty And Significance:** 2
**Empirical Novelty And Significance:** 2
**Recommendation:** 3
**Confidence:** 4

**Main Review:**

[Strengths]
1. Not only the audio synthesis but also two interesting applications (timbre transfer and signal compression) are demonstrated.

2. Audio samples are provided on the web page. They help understand the effectiveness of the proposed method.

[Weaknesses]
1. My primary concern is that the technical novelty is limited or not well described. Although the objectives are not entirely the same, a similar two-stage training scheme (in the first stage, only the reconstruction loss is used, and in the second stage, the adversarial loss is introduced) has already been used in previous studies (e.g., [A]). In Sections 3.1.1 and 3.1.2, I cannot find new losses or new techniques, which are unique to the presented task. At least, it would be better to clarify the technical difference from previous two-stage training schemes.

- [A] R. Yamamoto et al., “Parallel WaveGAN: A fast waveform generation model based on generative adversarial networks with multi-resolution spectrogram,” ICASSP 2020.

2. The combination of VAE and GAN has been proposed in previous studies, e.g., [B]. Also, in this respect, I feel that the technical novelty is limited. At least, it would be better to explain the relationships with such related work more politely.

- [B] A.B.L., Larsen et al., “Autoencoding beyond pixels using a learned similarity metric,” ICML 2016.

3. The improvement of synthesis speed is included in the main contribution. However, it mainly relies on incorporating multi-band decomposition, which was proposed in the previous study [Yu et al., 2019]. The combination of GAN-based waveform synthesis and multi-band decomposition was also already introduced in [C]. Therefore, I think that the incorporation of multi-band decomposition into the current framework is not so novel.

- [C] G. Yang et al., “Multi-band MelGAN: Faster Waveform Generation for High-Quality Text-to-Speech,” SLT 2021.

4. In Section 2, some related work is discussed. However, the discussion is minimal. For example, in Section 2.1, VAE is addressed under the title of “state-of-the-art”; however, only the original one [Kingma & Welling, 2014] and one variant [Higgings et al., 2016] are presented despite extensive studies in this field. In addition, there are many relevant work (e.g., [A][B][C]), which should be discussed more comprehensively in this paper.

5. In the experiments, 48 kHz audio waveforms are used for the proposed model, while 16 kHz audio waveforms are used for the baseline models. This experimental setting is unfair. I have a question (1) how performance is changed when the proposed model is applied to 16 kHz audio waveforms and (2) how performance is changed when the baseline models are applied to 48 kHz audio waveforms.

6. Although timbre transfer is an interesting application, it is not easy to judge whether the proposed model performs well. In general, the VAE encoder does not ensure that out-of-distribution data is encoded to the valid latent space. I cannot understand whether such an encoding error is alleviated in the proposed model.

**Summary Of The Paper:**

This paper aims to achieve fast and high-quality audio synthesis. The authors propose a variational autoencoder-based model called real-time audio variational autoencoder (RAVE) to achieve this aim. In particular, the authors introduce a two-stage training scheme to alleviate the difficulty in joint learning of the adequate representation and high-fidelity audio synthesis. More concretely, in the first stage, representation learning is conducted in a VAE framework. In the second stage, audio fidelity is improved using adversarial fine-tuning. The authors also introduce a post-analysis method to manipulate the balance between fidelity and compactness. The effectiveness of the proposed method was evaluated on the Strings and VCTK datasets. The applications to timbre transfer and signal compression were also demonstrated.

**Summary Of The Review:**

My primary concern is that the technical novelty is limited or not well described. The two-stage training scheme, the combination of VAE and GAN, and the incorporation of multi-band decomposition have already been proposed in the previous studies. I expect that the authors clarify this point in the rebuttal. In addition, I hope that the authors explain the validity of the experiments regarding the issues raised above.

---

> ### Author Response · Authors · 2021-11-17
> **Response to Reviewer Pcii**
>
> Thank you for your review. We've done our best to address your concerns in the latest paper version and in the comments below.
>
> > “My primary concern is that the technical novelty is limited or not well described. Although the objectives are not entirely the same, a similar two-stage training scheme (in the first stage, only the reconstruction loss is used, and in the second stage, the adversarial loss is introduced) has already been used in previous studies”
>
> The combination of a perceptual loss and an auxiliary loss have been used several times in previous approaches (Parallel Wavenet, Clarinet, Parallel WaveGAN, Multiband MelGAN), where the perceptual loss is used as a way to guide and stabilize the training process. In this context, starting to train those models only with the spectral loss can be seen as a way to implement warmup, which is not what we propose in RAVE. Our proposal is to use the perceptual loss (here the multiscale stft loss) during the representation learning stage in order to build a latent representation mirroring perceptual attributes of the input signal. It is only when the latent space converges to a stable state that we stop training the encoder, and start fine-tuning the decoder to produce higher quality audio signals. We have added a paragraph in the related work to make that difference clearer.
>
> However, our contribution is not limited to the aforementioned two-step training scheme. We also propose a post training analysis of the latent space allowing the user to balance between reconstruction fidelity and latent size compactness by estimating the dimensionality of the learned representation up to a given fidelity.
>
> > ​​”The combination of VAE and GAN has been proposed in previous studies”
>
> The key difference between previous approaches combining VAEs and GANs and RAVE is that we don’t train the encoder to optimize an adversarial loss. Initial experiments on RAVE have shown that training the encoder during the adversarial fine-tuning stage yields a larger estimated latent space dimensionality, which goes against our will to build a compact representation. We demonstrate it in the annex where we added a comparison between two trainings on the same dataset with and without freezing the encoder during the second stage. We have also added a paragraph in the related work to present the differences between our approach and the paper you mention.
>
> > “The improvement of synthesis speed is included in the main contribution. However, it mainly relies on incorporating multi-band decomposition, which was proposed in the previous study.”
>
> We are indeed building up from this previous study, and show that the multiband decomposition is a key element to 20 times faster than realtime neural audio synthesis on CPU at 48kHz, which, to the author’s best knowledge, has not been achieved yet.
>
> > “In Section 2, some related work is discussed. However, the discussion is minimal.”
>
> We added a related work section where we discuss several similar works with their key differences when compared to our approach.
>
> > “​​In the experiments, 48 kHz audio waveforms are used for the proposed model, while 16 kHz audio waveforms are used for the baseline models. This experimental setting is unfair.”
>
> During our experiments, we wanted to include a 48kHz NSynth model inside the perceptual test, but lacked the computational power and time to make it successfully converge, which led us to compare RAVE against the unmodified official implementations of both baselines.
>
> > “Although timbre transfer is an interesting application, it is not easy to judge whether the proposed model performs well. In general, the VAE encoder does not ensure that out-of-distribution data is encoded to the valid latent space.”
>
> This is a very interesting point, and we added an experiment to the annex to demonstrate the average KL divergence of the posterior distribution when encoding out-of-domain signals. We think that this problem might be addressed using an approach similar to [1, 2], where an adversarial loss is added to the encoder to make it domain agnostic. We however did not implement it inside RAVE, since our experiments have shown that it does perform timbre transfer without such mechanisms.
>
> [1] Lample et al. Fader Networks: Manipulating Images by Sliding Attributes, 2017
> [2] Mor et al. A Universal Music Translation Network, 2018

---

### Official Review · Reviewer_kQnp · 2021-11-03

**Correctness:** 3
**Technical Novelty And Significance:** 4
**Empirical Novelty And Significance:** 3
**Recommendation:** 8
**Confidence:** 4

**Main Review:**

Strength:
1. The synthesis speed is much faster than baselines due to non-autoregressive inference and the compact architecture.
2. The dimension of the latent representation can be significantly reduced without much degradation in audio quality. SVD is applied to omit less informative information in the latent representation.
3. The autoencoder is able to perform timbre transfer while largely preserving loudness and fundamental frequency.

Weakness:
It would be good to apply the proposed model as a vocoder in text-to-speech synthesis. For example, train a speech synthesis model to predict latent representation, which is then converted to waveform.

**Summary Of The Paper:**

This paper proposes an autoencoder for high-quality waveform synthesis. The approach is based on multi-band decomposition, VAE and adversarial fine-tuning. The synthesis speed is substantially faster than existing approaches on both CPU and GPU.

**Summary Of The Review:**

This paper proposes an audio synthesis approach that runs fast on CPU and yields high-fidelity audio. It combines VAE and GAN loss for non-autoregressive waveform synthesis. Although several use cases are discussed, no solid application (such as TTS) has been demonstrated.

---

> ### Author Response · Authors · 2021-11-17
> **Response to Reviewer kQnp**
>
> Thank you for your review and your encouraging comments! We agree that it would be very interesting to evaluate RAVE on a text-to-speech task. By proposing the autoregressive model trained on RAVE's prior, we hope to open the door to new ways of synthesizing audio: either as presented in the paper in a completely unconditional way, or conditioned on auxiliary signals (labels, acoustic descriptors, context...). Text to speech falls into this category of conditioned generation, but we believe that this lies outside the scope of this initial paper on audio representation learning.

---

### Author Response · Authors · 2021-11-17
**Updates**

We would like to thank all the reviewers for their time and expertise. In addition to the individual responses, we would like to clarify here the changes made to our article.

- New audio sample in the accompanying website
- Discuss key differences with previous approaches in the "Related work" section
- Added an experiment with out-of-domain data encoding (annex)
- Added an experiment to show how most of the latents are not informative (annex)
- Small changes and added references

---

### Decision · Program_Chairs · 2022-01-20

**Decision:**

Reject

**Comment:**

This paper presents an approach to high quality waveform synthesis using multi-band decomposition. The resulting synthesis speed is substantially faster the past work on both CPU and GPU -- a feature that all reviewers viewed as a significant strength. However, the majority of reviewers raised concerns about discussion of and contextualization within past work, as well the novelty of the proposed approach. Finally, one reviewer pointed out a potential concern with experimental evaluation (sample rate of proposed system outputs vs baseline's). Author response clarified the relationship with some past work but did provide evidence to mitigate the concerns about experimental evaluation. Overall, this paper could still benefit from another round of review.